# Activation of TF-Dependent Blood Coagulation Pathway and VEGF-A in Patients with Essential Thrombocythemia

**DOI:** 10.3390/medicina55020054

**Published:** 2019-02-16

**Authors:** Grażyna Gadomska, Katarzyna Ziołkowska, Joanna Boinska, Jan Filipiak, Danuta Rość

**Affiliations:** 1Department of Hematology and Malignant Diseases of Hematopoietic System, Faculty of Medicine, Nicolaus Copernicus University in Toruń, Collegium Medicum in Bydgoszcz, 85-168 Bydgoszcz, Poland; kasi10@wp.pl; 2Department of Pathophysiology, Faculty of Pharmacy, Nicolaus Copernicus University in Toruń, Collegium Medicum in Bydgoszcz, 85-094 Bydgoszcz, Poland; ziolkowska.ka@wp.pl (K.Z.); marlinforever@o2.pl (J.F.); joanna_jasiniewska@o2.pl (D.R.)

**Keywords:** blood coagulation, angiogenesis, tissue factor, vascular endothelial growth factor

## Abstract

*Background and objectives:* Recent studies suggest that a vascular endothelial growth factor (VEGF-A) may be involved in the thrombotic process by stimulating the expression of tissue factor in vascular endothelial cells. Tissue factor (TF) can also stimulate the transcription of the gene encoding VEGF-A. The relationship between coagulation and angiogenesis in myeloproliferative neoplasms is not fully understood. The aim of this study was to evaluate the concentration of TF in relation to VEGF-A in the blood of patients with essential thrombocythemia (ET). *Patients and methods:* The study group consisted of 130, newly diagnosed patients with ET (mean age 61 years). The control group consisted of 35 healthy volunteers (mean age 51 years). Concentrations of VEGF-A, TF, and tissue factor pathway inhibitor (TFPI) were analysed using immunoenzymatic methods. TF and TFPI activities were performed using chromogenic assays. *Results:* The median concentration of TF Ag was 3-fold higher and the TF activity was more than 15-fold higher in ET patients than in normal individuals. There were no statistically significant differences in the TFPI concentration and activity between groups. VEGF-A was significantly increased in patients with ET (*p* < 0.000001). Analysis of correlations revealed a positive correlation between VEGF-A and TF Ag as well as a positive correlation between VEGF-A and TFPI activity. *Conclusions:* The simultaneous increase of TF concentration and activity, VEGF-A in the blood of patients with ET, as well as a positive correlation between the concentration of TF and VEGF-A demonstrates the coexistence of TF-dependent coagulation and activation of angiogenesis.

## 1. Introduction

Essential thrombocythemia (ET) belongs to BCR-ABL negative myeloproliferative neoplasms, derived from multipotent hematopoietic progenitor cells. Approximately 55% of ET patients carry *JAK2* V617F mutation, and about 15–25% harbor *CARL* mutation and 4% *MPL* W515L/K mutation [1].

The clinical course of ET is associated with thrombotic complications relating to microvascular arteries and veins or bleeding disorders. The risk of thrombosis in patients with ET was estimated at 7.6–29.4% and bleedings at 3–18% [2]. Arterial thrombosis is more common than venous thrombosis and manifests as stroke, coronary heart disease, myocardial infarction, retinal artery thrombosis, and arterial thrombosis of lower extremities [3].

Many recent studies have focused on the pathogenetic mechanisms of the thrombotic process in ET patients. There is ongoing debate whether an increase in the number of blood cells (especially platelets) is solely responsible for thrombotic complications observed in the course of this disease [4,5]. Increased attention has recently been paid to the essential role of the extrinsic activation of plasma coagulation process triggered by tissue factor (TF). TF is released from damaged endothelial cells and activates monocytes, macrophages, leukocytes, and platelets. TF binds to the serine protease factor VII to form a complex. Factor VII is converted to the active form (TF-VIIa). TF-VIIa complex initiates a series of proteolytic events resulting in thrombin generation, which converts fibrinogen into fibrin [6,7,8].

TF is a glycoprotein released from damaged tissues, tumor cells, as well as endothelial cells and monocytes. It is activated by cytokines such as tumor necrosis factor, interleukin-6, and interleukin-8. Circulating leukocytes and activated platelets can also be a source of TF [9]. Cancer cells can constitutively release TF or may induce TF production by adjacent host cells including monocytes and endothelial cells [10]. Increased expression of TF was observed in head and neck cancers, prostate cancer, adenocarcinoma of the colon, and lung cancer. Tissue factor derived from tumor cells is not only involved in the clotting process but also plays an important role in many pathological processes such as angiogenesis and tumor metastasis.

Angiogenesis is crucial in the development of cancer. Although chaotic and highly abnormal, the vascular system inside the tumor provides oxygen and nutrients as well as determines tumor development and metastases. A factor of proven importance in the development of solid tumors, is a vascular endothelial growth factor (VEGF-A) [11,12].

Recent studies suggest that VEGF-A may be also involved in the thrombotic process through the stimulation of the expression of TF in vascular endothelial cells [11]. It has been proposed that TF can stimulate the transcription of the gene encoding VEGF-A [10,11,13]. The relationship between tissue factor and VEGF-A has been observed in breast cancer, small cell lung cancer, melanoma cells, and colorectal cancer [12,13]. Increased expression of TF in cancer cells has also been associated with increased VEGF-A levels and tumor size enlargement in mice [10].

The process of angiogenesis is also significant in the development of myeloproliferative neoplasms, which are characterised by enhanced microvessel density (MVD) [14,15,16]. Studies indicate that increased microvessel density positively correlates with VEGF-A concentration in patients with ET [12,14]. Therefore, high concentration of VEGF-A is considered an indicator of increased angiogenesis in ET patients [17,18].

The aim of this study was to evaluate the concentration of TF in relation to VEGF-A in the blood of patients with ET.

## 2. Patients and Methods

The study group consisted of 130, newly diagnosed patients with essential thrombocythemia (mean age 61 years; F/M = 84/46) and not treated with cytostatic drugs. Patients were recruited from the Department of Hematology and Malignant Diseases of Hematopoietic System, University Hospital No. 2 in Bydgoszcz, Poland. Complete blood count with peripheral blood smear, selected biochemical parameters, parameters of coagulation system, as well as bone marrow and cytogenetic analysis were performed in all patients with ET. ET was diagnosed according to the World Health Organization (2008) criteria.

Key exclusion criteria included other cancer and non-cancer diseases that can cause thrombocythemia, pregnancy, type 1 or type 2 diabetes, thrombotic or hemorrhagic complications at the time of diagnosis, stage III hypertension, as well as malignant hypertension.

The control group consisted of 35 healthy volunteers (mean age 51 years; F/M = 20/15).

The study was conducted during 2010–2015. The study was approved by the Bioethics Committee of Collegium Medicum in Bydgoszcz and the Nicolaus Copernicus University in Toruń. Written informed consent was obtained from all participants.

### 2.1. Material for Study and Methods

Blood samples were taken from an antecubital vein, after overnight fasting, to plastic tubes containing EDTA or 3.2% sodium citrate. Samples were centrifuged at 3000 rev/min for 20 min at 4 °C. The obtained plasma was divided into aliquots and stored at −80 °C until analysis but not longer than 6 months. Peripheral blood counts were performed on an Advia 120 hematology analyzer.

The following tests were performed using enzyme-linked immunosorbent assay (ELISA) methods: concentration of VEGF-A (R&D Systems Inc., Minneapolis, MN, USA), concentration of TF (IMUBIND^®^ Tissue Factor, Sekisui Diagnostics LLC., Stamford, CT, USA), and concentration of TFPI (IMUBIND^®^ TFPI ELISA kit, Sekisui Diagnostics LLC., Stamford, CT, USA). TF activity (ACTICHROME^®^TF, American Diagnostica Inc.) and TFPI activity (ACTICHROME^®^ TFPI Activity Assay, Sekisui Diagnostics LLC., Stamford, CT, USA) were performed using chromogenic assays. All measurements were made according to the manufacturer’s instructions.

### 2.2. Statistical Analysis

The statistical analysis was performed with the use of Statistica 12.0 software (StatStoft^®^ Cracow, Poland). The Shapiro-Wilk test was applied to assess the normality of distribution. Because our data did not meet the assumption of parametric statistics, the Mann-Whitney U test was employed for independent samples. Correlation coefficients were determined via the Spearman’s test. A *p*-value of less than 0.05 was considered statistically significant.

## 3. Results

Table 1 shows morphological parameters in patients with essential thrombocythemia and in the controls. A higher number of platelets and leukocytes were found in ET patients compared with the control group. There were no significant differences in erythrocytes count, hemoglobin concentration, and hematocrit between ET patients and controls.

Statistical analysis differences in concentration and activity of tissue factor in patients with ET compared with the control group were observed. The concentration of TF was significantly higher in patients with ET compared with the control group (Figure 1). The median concentration of TF Ag was 3-fold higher in ET patients than in the controls, and the activity of TF was more than 15-fold higher than in normal individuals (Figure 2).

Table 2 shows the concentration and activity of TFPI in patients with ET and in the control group. There were no statistically significant differences in the values of TFPI concentration and activity between groups. Figure 3 shows the concentration of VEGF-A in patients with ET and in the group of healthy individuals. VEGF-A was significantly increased in patients with ET (*p* < 0.000001) and the median in ET patients was more than five times higher than found in the control group (Me = 137.33 pg/mL versus Me = 23.00 pg/mL).

Analysis of correlations revealed a positive correlation between VEGF-A concentration and TF Ag (R = 0.27; *p* = 0.09) (Figure 4) as well as a positive correlation between VEGF-A concentration and TFPI activity in the group of ET patients (R = 0.26; *p* = 0.01) (Figure 5).

## 4. Discussion

The present study revealed a significantly higher concentration and activity of TF and VEGF-A in the blood of patients with ET compared with the control group. Increased expression of TF has also been reported in breast cancer, gastric cancer, lung cancer, colorectal cancer, prostate, and hematologic malignancies. Panova-Noeva et al. reported increased thrombin generation in patients with myeloproliferative neoplasms [19]. Falanga et al. observed significantly increased TF expression on platelets in patients with ET compared with controls [20]. Arellano-Rodrigo et al., described significantly enhanced expression of TF in ET patients with a history of thrombotic complications compared with ET patients without prior thrombotic events [21].

TF activation of the extrinsic coagulation pathway is regulated by the tissue factor pathway inhibitor (TFPI)—a natural inhibitor of blood coagulation. TFPI is combined with the coagulation factor Xa and inhibits TF/VIIa and Xa [6]. The role of TFPI in the pathogenesis of thrombosis in myeloproliferative neoplasms is not entirely clear.

In this study, there was no statistically significant difference in the concentration and activity of the TFPI in patients with ET compared with controls. Cacciola et al. observed a relationship between the concentration of TFPI and the thrombotic risk in patients with ET [22]. The research conducted by Bucallossi et al. showed decreased levels of TFPI caused by the acquired deficiency of different natural anticoagulants in patients with MPNs [17]. Falanga et al. have reported decreased levels of protein C and protein S in patients with ET and PV, which may lead to increased risk of thrombotic complications in these patients [18].

In this study, we have demonstrated significantly higher levels of VEGF-A in ET patients, which is consistent with reports by other authors [23,24]. A positive relationship between the concentration of VEGF-A and the concentration of TF was also found.

Experimental studies have shown that decreased TF expression is associated with tumor growth inhibition, diminished spread of tumor cells, and reduction of angiogenesis severity [9]. Clinical studies have shown a relationship between elevated levels of VEGF-A and increased concentration of TF as well as microvessel density in the bone marrow, which may confirm the role of tissue factor in angiogenesis. Tissue factor may stimulate the production of VEGF-A, which also can increase the synthesis of TF on endothelial cells. Nakasaki et al. demonstrated in patients with colorectal cancer that the expression of VEGF-A and TF is dependent on hypoxia or anoxia. However, increased synthesis of both of these factors may be due to various mechanisms. Experimental studies have indicated that VEGF-A production is stimulated by hypoxia-inducible transcription factor 1 (HIF-1), and TF synthesis is carried out by independent HIF [11,25].

TF affects proangiogenic phenotype of the cells by increasing the expression of VEGF-A through a paracrine signaling PAR-1; it also reduces antiangiogenic proteins like thrombospondin [26,27]. TF through PAR-1 receptors can stimulate angiogenesis [28]. This has been confirmed by animal studies that have shown the death of embryos lacking PAR-1 to be caused by inhibited growth of blood vessels [29]. In addition, research has indicated a strong mitogenic effect of PAR-1 on the endothelial cells and extracellular matrix synthesis [30].

In the present study, a positive relationship between the concentration of VEGF-A and the activity of TFPI in patients with ET was observed. Increased TFPI activity with higher concentration of TF may reflect a compensatory mechanism for procoagulant activity in ET patients.

## 5. Conclusions

The simultaneous increase of TF concentration and activity as well as the presence of VEGF-A in the blood of patients with ET, in addition to a positive correlation between the concentration of TF and VEGF-A, demonstrates the coexistence of TF-dependent coagulation and activation of angiogenesis.

## Figures and Tables

**Figure 1 medicina-55-00054-f001:**
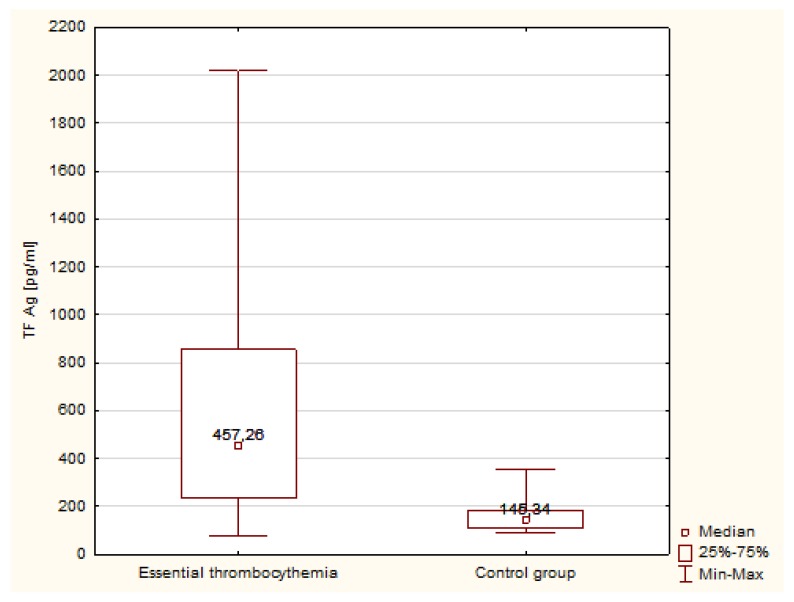
Concentration of TF in patients with ET and in the control group.

**Figure 2 medicina-55-00054-f002:**
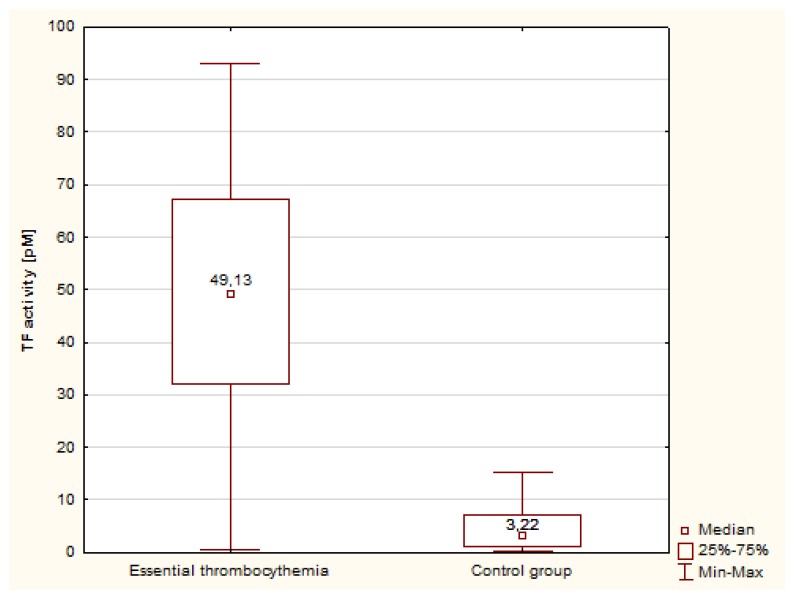
Concentration of TF activity in patients with ET and in the control group.

**Figure 3 medicina-55-00054-f003:**
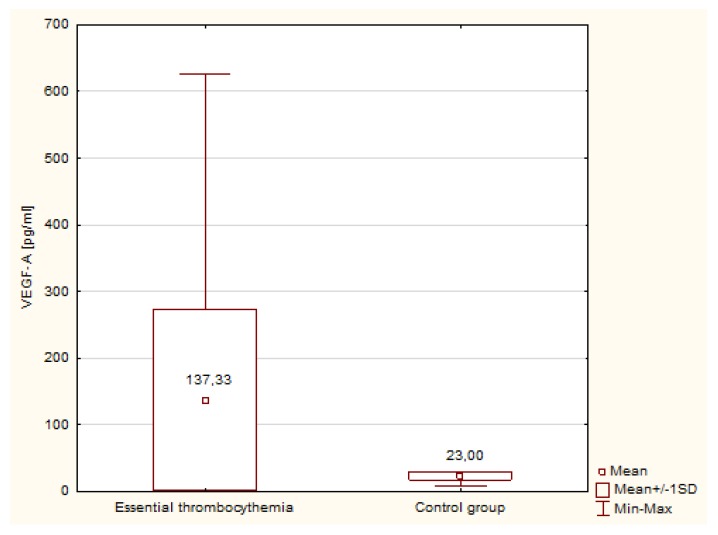
Concentration of VEGF-A in patients with ET and in the control group.

**Figure 4 medicina-55-00054-f004:**
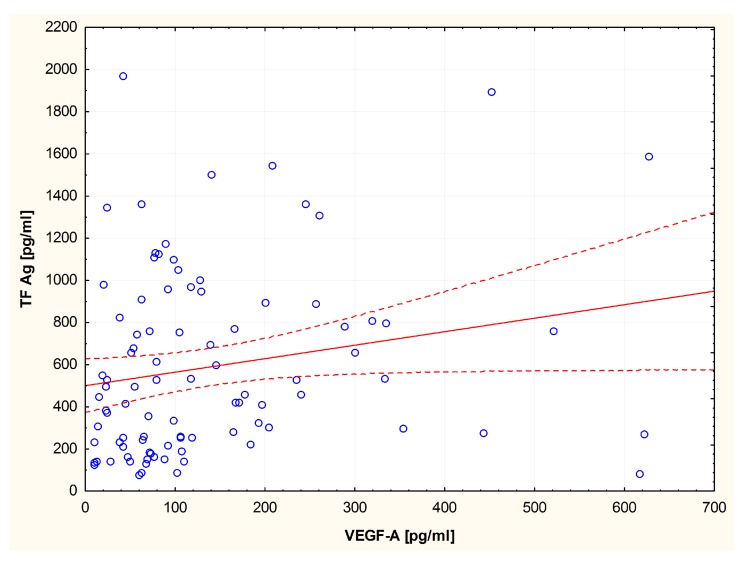
Spearman’s correlation between VEGF-A and TF Ag in patients with ET.

**Figure 5 medicina-55-00054-f005:**
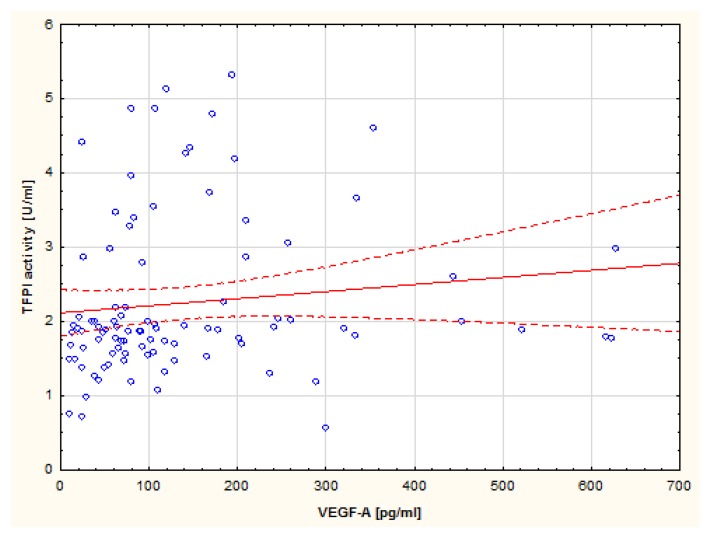
Spearman’s correlation between VEGF-A and TFPI activity in patients with ET.

**Table 1 medicina-55-00054-t001:** Morphological parameters in patients with essential thrombocythemia and in the control group.

Parameters	Essential Thrombocythemia *N* = 130	Control Group *N* = 35	*p*
	Me (Q1;Q3)	Me (Q1;Q3)	
Erythrocytes, T/L	4.99 (4.62;5.43)	4.86 (4.42;5.22)	NS
Hemoglobin, g/dL	14.55 (13.70;15.40)	13.80 (13.00;14.60)	NS
Hematocrit, %	44.15 (41.55;46.70)	44.20 (40.60;46.40)	NS
Leukocytes, G/L	9.72 (6.24;11,34)	6.00 (5,40;6,90)	<0.001
Platelets, G/L	811.00 (664.00;947.00)	256.00 (223.00;287.00)	<0.001

NS = not significant. Mann-Whitney-U test was performed to compare results between the two groups.

**Table 2 medicina-55-00054-t002:** Concentration and activity of TFPI in patients with ET and in the control group.

Parameters	Essential Thrombocythemia *N* = 130		Control Group *N* = 35		*p*
	Me (Q1;Q3)	Min.–Max.	Me (Q1;Q3)	Min.–Max.	
TFPI Ag, ng/mL	82.52 (64.48;111.32)	28.48–235.72	81.92 (62.82;90.24)	52.22–114.68	NS
TFPI activity, U/mL	1.89 (1.64;2.80)	0.56–5.32	1.88 (1.15;2.49)	0.80–3.98	NS

Ag—antigen; NS—not significant. Mann-Whitney-U test was performed to compare results between the two groups.

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
