# Peer review of "Activation of TF-Dependent Blood Coagulation Pathway and VEGF-A in Patients with Essential Thrombocythemia"

_medicina, 2019, doi:10.3390/medicina55020054_

Round 1
Reviewer 1 Report
This work by Gadomska G. et al. reports about the expression of TF and VEGF-A in Essential Thrombocythemia. The authors have published previously on this topic, and I would have preferred to see also the citation of their works among the references. In fact, the paper from Gadomska G et al. published in 2016 in Blood Coagul. Fibrolysis is not reported. Some other references are lacking, for instance at lane 66 authors should add a reference on the transcriptional regulation of VEGF-A by TF. A reference on the inhibition of VEGF-A in myeloproliferative neoplasms is also needed.
In the Methods section, table should not be present; they should be discussed in the Result section. Within the Methods, a paragraph on the statistical evaluation of the data should be added.
Authors have no data about changing in TF or VEGF-A expression in patient serum following therapeutic intervention?
Minor points:
1) VEGF should be converted into VEGF-A;
2) Abbreviations must be verified, at least the first time a protein is mentioned it should be written in full length followed by the abbreviation. Then, only the abbreviated form should be used;
3) Lane 15: the transcription of the gene encoding VEGF-A can be augmented, increased, up-regulated, but not grown;
4) Lane 17 and thereafter: it is important to say if TF concentration is evaluated at the mRNA or protein level or both;
5) Lane 21: there is not only one immunoenzymatic method, to which one the authors refer to?
6) The sentences at lane 44-51 are not clear.
7) The country of the providers of the enzymatic assays should be added. Which protocol has been used? The one suggested by the manufacturers?
Various typing errors are present in the text. I.e.:
Lane 13. Introduction
Lane 19. Thrombocythemia
Lane 41: manifests
Lane 82-84: check the character size
Author Response
We hope, that the paper after revision will be suitable for publication.

Reviewer 2 Report
The manuscript describes about study to evaluate the concentration of TF in relation of VEGF-A in the blood of patients with essential thrombocythemia. The manuscript is well presented and conclusion is supported by various studies and data. Following are some comments which needs to be addressed by authors before publication.
Decision: Minor Revision
Comments:
1. Methods used (ELISA) and if other should be briefly described in manuscript.
2. Manuscript needs to be revised for sentence, grammar and punctuation errors.
3. Authors should add, model and make of equipments used in all experiments
4. Authors should also give information about which statistical methods have been utilized.
Author Response

(The authors gave the same response as above.)

Round 2
Reviewer 1 Report
The authors reviewed the manuscript according to my suggestions.
Minor things:
1) Lane 15: please, change the sentence into: "TF can also stimulate the transcription of the gene.....".
2) Lane 21: in a similar way, change the sentence into: "Concentration of VEGF-A as well as concentrations and activities of TF and tissue factor pathway inhibitor (TFPI) were analysed using an immunoenzymatic method".
3) Lane 26: is this statement sure "Analysis of correlations revealed a positive correlation between VEGF-A and TF Ag as well as a positive correlation between VEGF-A and TFPI activity", or the positive correlation is between VEGF-A and TF activity? Same thing at lane 146, 152 and 195. This result on TFPI activity is also lacking from the conclusions. This aspect should be explained better.
4) Lane 35 and lane 47: please, repeat the extended name and abbreviation here for ET and TF respectively, even if it has been already made in the abstract. Conversely, insert the abbreviation from here (see lanes 52, 55, 65, 67 and others where the extended name is still present).
5) Lane 66: change this sentence into: "stimulate the transcription of the gene encoding VEGF-A".
6) Paragraph 2.1: could you please separate the activity assays from the ELISAs and explain on which principle the activity assays are based?
7) Lane 148 and lane 152: change "liner" into "linear".
8) Lane 183: change the sentence as follows: "Nakasaki et al. demonstrated in patients with colorectal cancer that the expression of VEGF-A and TF is dependent on hypoxia or anoxia. However, increased synthesis of both of these factors may be due to various mechanisms. Experimental studies.......".
9) Lane 194: this last sentence of the discussion is important, it should be improved and amplified.
Author Response
The authors wish to thank the Reviewer 2. All of the minor mistakes has been corrected according to the Reviewer suggestions.
A positive relationship between the concentration of VEGF-A and TFPI activity in ET patients probably reflect compensatory mechanism for procoagulant activity observed in ET patients. However, median TFPI Ag and activity in ET patients were similar to observed in the control group. For this reason, authors decided to leave the conclusions unchanged.
We hope, that the paper after revision will be suitable for publication.